# Modulating the Viscoelastic Properties of Covalently Crosslinked Protein Hydrogels

**DOI:** 10.3390/gels9060481

**Published:** 2023-06-12

**Authors:** Rossana Boni, Lynne Regan

**Affiliations:** Centre for Engineering Biology, School of Biological Sciences, Institute of Quantitative Biology, Biochemistry and Biotechnology, University of Edinburgh, Edinburgh EH9 3FF, UK

**Keywords:** hydrogels, protein engineering, biomaterials, SpyTag–SpyCatcher

## Abstract

Protein engineering allows for the programming of specific building blocks to form functional and novel materials with customisable physical properties suitable for tailored engineering applications. We have successfully designed and programmed engineered proteins to form covalent molecular networks with defined physical characteristics. Our hydrogel design incorporates the SpyTag (ST) peptide and SpyCatcher (SC) protein that spontaneously form covalent crosslinks upon mixing. This genetically encodable chemistry allowed us to easily incorporate two stiff and rod-like recombinant proteins in the hydrogels and modulate the resulting viscoelastic properties. We demonstrated how differences in the composition of the microscopic building blocks change the macroscopic viscoelastic properties of the hydrogels. We specifically investigated how the identity of the protein pairs, the molar ratio of ST:SC, and the concentration of the proteins influence the viscoelastic response of the hydrogels. By showing tuneable changes in protein hydrogel rheology, we increased the capabilities of synthetic biology to create novel materials, allowing engineering biology to interface with soft matter, tissue engineering, and material science.

## 1. Introduction

Hydrogels are an important class of materials widely used as adsorbents, drug delivery depots, biosensors, and microfluidics devices [1,2,3]. Depending on the nature of the application, the physical properties of the hydrogels must be controlled, fine-tuned, and customised. For example, hydrogels used for the regeneration of tissue must mimic the physical and biological properties of the native tissue, prompting the field of tissue engineering to pivot towards the development of biological rather than synthetic hydrogels due to their biocompatibility and tunability [4,5]. Recently, hydrogels composed of recombinant proteins have gained interest due to the precise control over the structure and features offered by protein engineering. In particular, using amino acids as building blocks allows one to encode the desired hydrogel features in the protein sequence that specifies the structure, achieving great customisation in the macroscopic hydrogels [6]. Our design is based on protein-only hydrogels that are covalently crosslinked using the SpyTag–SpyCatcher system. The SpyTag–SpyCatcher system was developed from a bacterial protein that naturally forms an intramolecular covalent bond, the CnaB2 domain of the FbaB protein from *Streptococcus pyogenes* [7]. Howarth and co-workers were able to split the protein in two, creating the two reactive protein partners SpyTag (ST) and SpyCatcher (SC), whilst maintaining their ability to form the covalent bond [7]. What makes this system particularly unique is that the covalent bond between ST and SC forms spontaneously upon mixing under mild conditions, such as aqueous media and pH 7 [7]. We have shown that ST and SC can be successfully used to form biocompatible covalent molecular networks with distinct physical properties that allow for the encapsulation of mammalian cells without loss of cell viability [8]. Herein, we focussed on the systematic characterisation of how the rheological properties of the protein hydrogels change depending on the ratio of ST:SC and the protein concentration. Understanding the relationship between the chemical structure and rheology of protein hydrogels is crucial to master the versatility of the system and tailor the recombinant production of the protein precursors to suit a specific application. We elected to exploit the use of a chemically crosslinked system based on the recreation of the covalent bond between ST and SC, as opposed to physically crosslinked systems where the polymer chains entangle via transient and reversible junctions [9]. Covalently crosslinked hydrogels are characterised by stronger mechanical properties and irreversible bonds. These characteristics translate into different rheological behaviours observable in frequency sweeps. In particular, physically crosslinked systems will only present suitable viscoelastic properties within a short timescale (i.e., high frequencies) due to the reversibility of their bonds, whilst chemically crosslinked hydrogels will present the same viscoelastic properties towards infinitely low frequencies, indicative of the stability of their bonds [9]. There are plenty of accounts in the literature investigating the rheological properties of synthetic polymer-based materials or naturally occurring protein hydrogels, such as collagen or agarose [10,11]. However, to the best of our knowledge, there are no accurate accounts detailing the changes in viscoelastic behaviour of chemically crosslinked hydrogels based on recombinant proteins. Given the recent advances in protein engineering and the consequent interest in recombinant protein hydrogels, we sought to determine how the rheological properties of recombinant hydrogels change based on the protein concentration and the ratio between the components. Our approach to hydrogel design, development, and optimisation is not only straightforward, as ST–SC forms a hydrogel at room temperature without the need for an additional chemical crosslinker, but also clear and uncomplicated, as we demonstrate how the rheological properties of the hydrogels can be modified only by rationally adding or removing the protein components. The viscoelastic behaviour of hydrogels investigated via bulk rheology is crucial to determine the suitability of protein-based hydrogels for tissue engineering applications, but, most importantly, the rational and direct modification of the viscoelastic properties of protein-based hydrogels is an advantageous predictive tool to move towards the identification of proteins with the desired physical properties, such as mechanical stability and elasticity, that could be able to expand the scope of the SpyTag–SpyCatcher system beyond this study and increase the sophistication of protein-based biomaterials.

## 2. Results and Discussion

### 2.1. The Protein Building Blocks

*S. aureus* surface protein G, SasG, is an elongated, stiff, rod-like protein formed by tandem repeats of two structurally related domains: E (50 residues) and G (78 residues) [12]. The structure of SasG (GEG) has been previously determined by X-ray crystallography (PDB ID: 3TIQ) and longer SasG arrays, composed of up to 7 EG repeats following the core monomer GEG, have been characterised in solution by small angle X-ray light scattering (SAXS) and atomic force microscopy (AFM) [13]. In this study, we used SasG composed of the core monomer GEG as a crosslinker to form strong hydrogels. We also created a longer version of the crosslinker, which we named SasGlong, by combining the core SasG monomer GEG with three tandem repeats of EG, resulting in ((GEG) + 3x(EG)). Previous physical characterisation via SAXS and AFM indicated that the longer version of SasG, SasGlong, shares its rod-shaped, elongated, and stiff properties [13]. Both SasG and SasGlong were engineered to carry two SpyTag (ST) motifs, one at each end of the chain, resulting in two constructs: ST–SasG–ST and ST–SasGlong–ST (Figure 1A). The final length of each construct was ~17 nm end-to-end for SasG and ~51.5 nm end-to-end for SasGlong. The SC arrays were engineered by fusing three or four SC units together via a flexible glycine rich linker, (GGS)_2_RS (Figure 1A). Our previous SAXS characterisation showed that the SC arrays behave like extended flexible structures, comparable to beads on a string, with no evidence of intra- or inter-chain aggregation [8]. Upon mixing at room temperature, a covalent cross-link spontaneously forms between dissolved ST and SC, resulting in a percolated network (Figure 1B). The quick and spontaneous formation of the hydrogel network is the first advantage of the crosslinking method we chose. Indeed, whilst chemical crosslinking methods are usually characterised by strong mechanical properties, they usually require the use of a chemical crosslinking agent, such as copper for the click-chemistry copper-catalysed alkyne–azide cycloaddition reaction, that hinders the biocompatibility and the efficiency of the reaction [9]. Instead, the ST–SC reaction does not require the addition of a chemical crosslink, as the covalent bond between the unprotonated lysine of SpyCatcher and the aspartic acid in SpyTag is simply catalysed by the neighbouring glutamic acid [7]. Moreover, the ST–SC reaction has been shown to be stable in a wide variety of conditions, including temperatures, pHs, and buffers [7], and we did not observe any sensitivity to any of the aforementioned conditions during hydrogel preparation. Notably, the proteins involved in the ST–SC reaction must be folded, as protein denaturation with urea after gelation leads to a change in the viscoelastic properties of the resulting hydrogels [8]. This simple gelation system has already been exploited elsewhere to encapsule mammalian cells thanks to the uncomplicated gelation kinetics [14,15]. Here, we sought to investigate further how the ratio of ST–SC units and the total protein concentration modulate the viscoelastic properties of the resulting hydrogels. In particular, we wanted to determine the suitability of ST–SC hydrogels as scaffolds for mammalian cell growth as a bioink for 3D printing applications. For the former, it has been clearly established that for successful encapsulation and growth of cells onto a matrix, that matrix has to mimic the viscoelastic properties of the native tissue [16], demonstrating how the viscoelastic properties of our agnostic ST–SC hydrogels are crucial for tailoring them to different organs characterised by different viscoelasticity, i.e., liver (10,000 Pa) or brain (1000 Pa) [17]. For the second requirement, 3D printing, the resulting viscoelastic properties and the speed of gelation of our ST–SC hydrogels were crucial to determine the right combination to use. For instance, for 3D printing, we needed a combination that did not gel too quickly, as this would have led to the clogging up of the nozzle and unsuccessful 3D printing, whilst also gelling fast enough to maintain its structural integrity whilst the second layer was printed on top. Moreover, we needed to ascertain that the resulting viscoelastic properties were still suitable for tissue engineering applications. Before investigating the rheological properties of our hydrogels, we investigated their swelling properties to determine if changes in the molar concentrations led to changes in the swelling behaviour. We determined this not to be the case, as all of our hydrogels showed similar swelling of ~50% after 24 h (Appendix A).

### 2.2. Viscoelastic Properties of the Protein Hydrogels

Naming convention: we will refer to the composition of a hydrogel based on the molar ratio of ST to SC. For example, a hydrogel made from ST–SasG–ST at 2 mM and SC3 at 1 mM has 4 moles of ST (2 mM × (2 SpyTags in each SasG chain)) and 3 moles of SC (1 mM × 3 SC units); therefore, the network is defined as ST:SC 1.3:1. Table 1 shows the composition and names of the hydrogels described in this paper.

Figure 2 summarises our findings and illustrates the trends in the viscoelastic properties of the hydrogels. Regardless of the proteins used to make the hydrogels, when SC is present in great molar excess compared to ST, the mixture results in a viscous liquid, rather than a hydrogel (G′ < G″). Conversely, when the ST component is present in great molar excess compared to SC, a hydrogel is formed with G′ = 1000 Pa, which does not yield to 100% strain. When ST and SC are approximately equimolar, the resulting hydrogels exhibit G′ = 10,000 Pa and critical yield stress at 10%. As the total concentration of equimolar ST: SC is increased, a hydrogel with G′ = 1000 which does not yield at 100% strain is formed.

During the rheological investigation of the protein hydrogels, we observed that the identity of the protein components, the molar ratio between the ST and SC units, and the total protein concentration were all determining factors in the resulting rheological properties. Below, we discuss this behaviour in detail for each variable.

### 2.3. Effects of the ST:SC Ratio on the Viscoelastic Properties of the Resulting Hydrogels

#### 2.3.1. SC3 Combined with ST–SasG–ST and ST–SasGlong–ST

The combination of ST–SasG–ST:SC3 0.6:1 showed G′ < G″, indicating that the combination remained a viscous liquid. ST–SasG–ST:SC3 1.3:1 showed G′ = 10,000 Pa, G′ > G″, and G′ independent of frequency. This hydrogel showed critical yield stress at 10% strain. Moreover, the hydrogels formed at all tested ratios between 1.3:1 and 2.6:1 showed G′ of 1000 Pa with G′ > G″, and G′ was independent of frequency. These gels did not yield up to 100% strain (Figure 3). Combinations of SC3 with ST–SasGlong–ST, at all ratios and concentrations analysed, exhibited G′ = 10,000 Pa and critical yield stress at 10% strain (Appendix A).

#### 2.3.2. SC4 Combined with ST–SasG–ST and ST–SasGlong–ST

The combination of ST–SasG–ST:SC4 0.5:1 showed G′ < G″, indicating the permanence of a viscous liquid. The hydrogel formed by ST–SasG–ST:SC4 1:1 showed G′ = 10,000 Pa and yielded at 10% strain, whereas at all molar ratios beyond ST:SC 1:1, up to 2:1, the hydrogels exhibited G′ = 1000 Pa, G′ > G″, and no yield up until 100% strain (Appendix A). ST–SasGlong–ST plus SC4 at 1:1 resulted in a gel with G′ = 10,000 Pa and yield at 10% strain, and at 2:1 resulted in a gel with G′ = 1000 Pa and no yield up to 100% strain (Appendix A). Therefore, ST–SasG–ST and ST–SasGlong–ST behaved similarly when combined with SC4. ST–SasG–ST combined with SC3 also followed this behaviour, but ST–SasGlong–ST combined with SC3 deviated from the observed trend.

#### 2.3.3. Effects of the Total Protein Concentration on the Viscoelastic Properties of the Resulting Hydrogels

We investigated the dependence of the viscoelastic properties on total protein concentration by performing strain and frequency sweeps on hydrogels where the ST:SC ratio was kept constant at 1:1, but the total protein concentration was increased 1.5× at each step. Appendix A details the specific concentrations analysed.

#### 2.3.4. SC3 Combined with ST–SasG–ST and ST–SasGlong–ST

For SC3 and ST–SasG–ST, the frequency sweeps showed that at low mM concentrations, G′ < G″, indicative of viscous liquid behaviour (Appendix A). At a higher concentration of SC3 (1.6 mM) and ST–SasG–ST (2.5 mM), the combination formed a hydrogel characterised by G′ > G″, G′ = 10,000 Pa, and yielding at 10% strain. Further increase in the total protein concentration of both SC3 (2.5 mM) and ST–SasG–ST (3.7 mM) led to the formation of a hydrogel with G′ = 1000 Pa and no yielding until 100% strain (Figure 4A). By contrast, in hydrogels formed by SC3 and ST–SasGlong–ST, the physical properties did not change (G′ = 10,000 Pa and critical yield stress at 10% strain) despite increasing the protein concentration (Appendix A).

#### 2.3.5. SC4 Combined with ST–SasG–ST and ST–SasGlong–ST

Similarly, for SC4 and ST–SasG–ST, the frequency sweeps showed the permanence of a viscous liquid, G′ < G″, at low mM concentrations (Appendix A). By increasing the concentration of ST–SasG–ST + SC4 to 2.5 mM + 1.2 mM and 3.7 mM + 1.8 mM, respectively, the resulting hydrogels exhibited G′ = 1000 Pa and no yielding up to 100% strain (Figure 4B). Hydrogels formed by combinations of ST–SasGlong–ST (2.5 mM) and SC4 (1.2 mM) showed G′ = 10,000 Pa and critical yield stress at 10% strain, whilst hydrogels formed by ST–SasGlong–ST (3.7 mM) and SC4 (1.8 mM) exhibited G′ = 1000 Pa and no yielding up to 100% strain (Appendix A).

The importance of protein-based hydrogels in tissue engineering, biophysics, and soft matter is steadily growing due to the precise control offered by protein engineering over the resulting hydrogel properties. Herein, we present the results of systematically investigating how the properties of the hydrogels can be modulated in response to changes in the ratio of the components and the total protein concentration.

By varying the ratio between the ST and SC units, we found that when the SC units are in excess of the ST units, a viscous liquid results and a gel is not formed. We rationalise this result with an excess in protein binding sites (SCs) available, where the majority of the protein–protein interactions will not lead to productive crosslinking and the formation of a percolating network. Intuitively, when SC and ST units are mixed in approximately equimolar concentrations and at a relatively low concentration, the strongest gel forms, with G′ = 10,000 and critical yield stress at 10% strain. This is because we maximised crosslinking between ST and SC, and no extra moieties influence the behaviour of the system. Finally, when ST is present in excess compared to SC, the hydrogels weaken (G′ = 1000 Pa), but the resistance to deformation is increased (no yielding up until 100% strain). We ascribed this phenomenon to an excess in ST, that yields a heterogeneous network. Others have reported similar phenomena, where protein hydrogels do not form (G′ < G″) when one component is present in great excess compared to another, as well as a sharp decrease in elastic modulus when the ratio between components is unbalanced [6,18,19]. Moreover, it has been previously shown that steric hinderances caused by large chains induce the formation of weak gels and further increases in steric congestion hinder their formation completely [20,21]. Therefore, we propose that our ST–SC system presents the same characteristics as the ones highlighted before, where steric hindrances result in some of the crosslinking units (the ST–SasG–ST, in our case) not being available to form the second cross-link after the first has formed or binding only one SC array, creating a heterogenous network. Therefore, an excess of ST units over SC units leads to the formation of weaker hydrogels. The same steric hindrances affect the hydrogels characterised by higher total protein concentration with equimolar concentrations of SC and ST units (G′ = 1000 Pa and no yielding up to 100% strain). In this case, the increased viscosity acts as the limiting factor, weaking the hydrogels as some crosslinking units are unable to form a covalent bond [21]. Of note, we carried out additional experiments above the 4 mM concentration (ST–SasG–ST = 6 mM), but we realized that the viscosity of the solution was too high, and the viscoelastic properties of the hydrogels were further reduced. Therefore, we did not present any data above 4 mM as we believe this to be the threshold above which the mechanical properties of the hydrogels are extremely limited.

We also investigated the gelation kinetics of the protein hydrogels using microrheology and identified that the 1:1 combination had relative slow kinetics (complete gelation was achieved in ~45 min), whilst increasing the ST units led to progressively faster gelation kinetics until complete gelation was recorded in less than 10 min (Appendix A). A similar behaviour was also reported elsewhere [6,21,22], demonstrating how the speed of gelation influences the macroscopic mechanical properties of the hydrogels. Interestingly, the combinations of ST–SasGlong–ST and SC3 showed slower and incomplete gelation, supporting the idea of underlying steric hinderances limiting the formation of a percolating network (Appendix A).

In addition, we speculate that lengthening the SC array lowers the ST–SC threshold needed to form a hydrogel, in terms of both the ST:SC ratio and protein concentration. This can already be seen in Figure 2, where at the same 1:1 concentration, ST–SasG–ST exhibited G′ = 10,000 Pa when combined with SC3, but G′ = 1000 Pa when combined with SC4. The same can also be observed for combinations of ST–SasGlong–ST with SC3 or SC4. Therefore, we predict that further lengthening the SC array by genetically engineering a longer version of the protein could push the formation of a hydrogel at a very low protein concentration. However, we acknowledge that it is not trivial to genetically engineer longer versions of the SC arrays due to their very repetitive sequence. Finally, the G′ of these networks are in good accordance with those of medium viscoelastic organs, such as livers and kidneys, highlighting the suitability of these hydrogels for tissue engineering applications [17].

## 3. Conclusions

In summary, the rheological results presented here offer a clear blueprint of rheological behaviour for protein hydrogel applications. We determined the molar ratio and the size of the proteins needed in order to create a hydrogel with the desired characteristics. For example, an excess of ST compared to SC will result in a weak gel structure exhibiting high yield stress, whilst the opposite combination is unlikely to form a self-sustaining network. We also showed that increasing the protein concentration of the hydrogels at the same ST:SC ratio showed a similar behaviour, where the increased viscosity acts as a limiting factor, weaking the hydrogels. Finally, we determined that the rheological properties of the protein hydrogels are suitable for tissue engineering applications.

## 4. Materials and Methods

### 4.1. Bacterial Strains, Plasmids, Culture Conditions

The bacterial strains and plasmids used in this study are listed in the Appendix A.

### 4.2. Recombinant Protein Expression and Purification

*E. coli* cells harbouring the appropriate plasmid were grown at 37 °C with 250 rpm shaking in Luria Bertani broth to an optical density of 0.6–0.8 at 600 nm. Protein expression was induced with 1 mM IPTG and growth continued for a further 20 h at 18 °C. Cells were harvested and collected by centrifugation at 10,000 rpm for 10 min and pellets were stored at −20 °C until needed. The His-tagged proteins were purified from the frozen cells using ion metal affinity chromatography (Ni-NTA resin; Qiagen, China) using a batch method. Briefly, after cell disruption, whole cell lysate was incubated with 8 mL of Ni-NTA resin for an hour at 4 °C on a shaking platform. The resin was washed twice with Tris–HCl, Imidazole 20 mM buffer, pH 8, and the protein of interest was eluted using Tris–HCl, Imidazole 200 mM buffer, pH 8. Protein expression and purification were assessed by SDS–PAGE. The purified protein was then dialysed extensively against distilled water at 4 °C for 12 h, frozen at −80 °C, and then lyophilised. Lyophilised proteins were stored at −80 °C until use. Protein purification was verified via Comassie Blue staining on an SDS-PAGE. We observed no significant variation between batches.

### 4.3. Preparation of the ST–SC Hydrogels

Lyophilised proteins were individually dissolved in distilled water to give stock solutions of the desired concentration. The appropriate concentrations were determined via A280 and the extinction coefficient calculated from the sequence (detailed in the Appendix A). The ST and SC components were mixed at room temperature by gentle pipetting at the predetermined molar ratio. Gelation occurred spontaneously upon mixing.

### 4.4. Water Intake

A combination of 30 µL of ST–SasG–ST and 30 µL of SC at the appropriate mM concentrations were manually mixed together to initiate spontaneous gelation. After gelation was complete, 1 mL of ultrapure water was added to each Eppendorf tube containing a hydrogel, and the mixture was incubated overnight at room temperature. Excess water was removed, and the hydrogels were weighed (W_w_). Subsequently, the hydrogels were freeze-dried and weighed again (W_0_). The percent water intake was calculated using: (W_w_ − W_0_) × 100/W_0_.

### 4.5. Dynamic Shear Rheology and Data Analysis

Rheological measurements were carried out using a stress-controlled Discovery Hybrid Rheometer DHR-2 (TA Instruments, New Castle, DE, USA) with a standard steel parallel plate geometry (8 mm diameter). The linear viscoelastic region (LVR) was determined via a strain sweep with strain amplitude increasing from 0.01 to 100% and a frequency of 100 rad/s. Following each strain sweep, frequency sweeps were carried out in the established LVR by holding the strain at 1% and decreasing the oscillatory frequency ω from 100 to 0.1 rad/sec. The storage (G′) and loss moduli (G″) were determined as a function of ω at 25 °C. Three independent measurements were recorded, and the mean is reported. Graphs were made using Prism 9 for MacOS, version 9.3.1.

## Figures and Tables

**Figure 1 gels-09-00481-f001:**
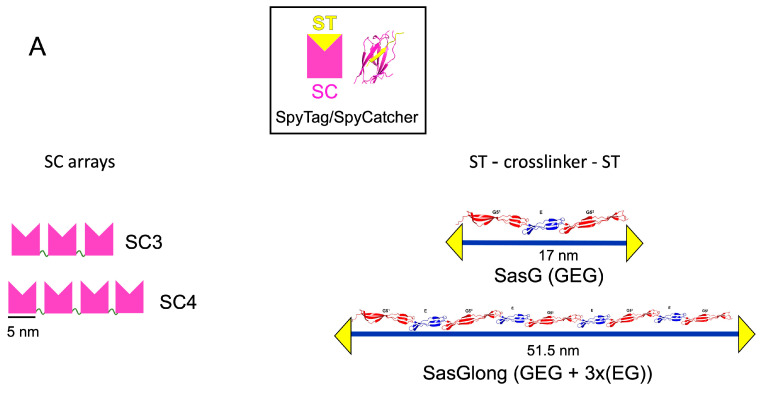
Schematic illustration of the components of the SpyTag (ST)—SpyCatcher (SC) hydrogels. (**A**) Top: Ribbon representation of the ST–SC complex, with the ‘yellow triangle and pink crown’ cartoon used to indicate ST and SC, respectively. Left: pink crowns depicting the SC arrays linked by glycine-rich flexible linkers (green). Right: ribbon representation and cartoon of the two crosslinkers SasG (GEG) and SasGlong (GEG + 3x(EG); blue line) with a single ST at the terminals. (**B**) Schematic representation of the crosslinking between the SC3 array and ST–SasG–ST, resulting in a covalently crosslinked hydrogel.

**Figure 2 gels-09-00481-f002:**
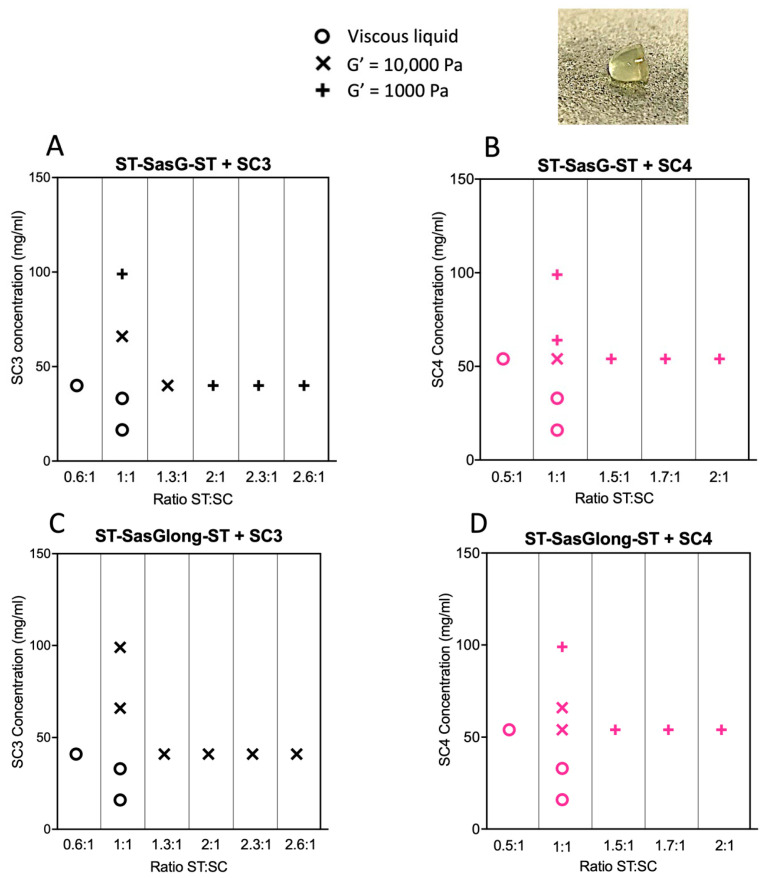
Hydrogel formation and viscoelastic properties of the ST–SC protein hydrogels. (**A**) ST–SasG–ST combined with SC3 (black); (**B**) ST–SasG–ST combined with SC4 (pink); (**C**) ST–SasGlong–ST combined with SC3 (black); (**D**) ST–SasGlong–ST combined with SC4 (pink). The insert shows an example of the self-standing mouldable protein hydrogels we developed. The hydrogel was made from 30 µL of ST–SasG–ST at 4 mM and 30 µL of SC3 at 1 mM.

**Figure 3 gels-09-00481-f003:**
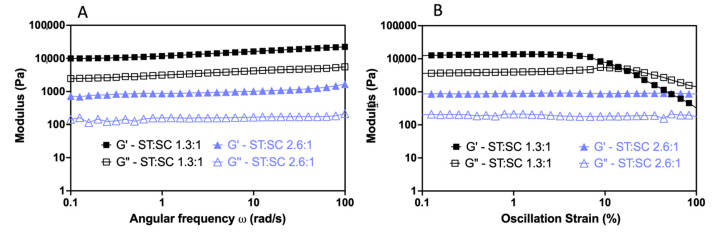
Viscoelastic changes in response to ST–SasG–ST and SC3 ratio variation in frequency (**A**) and strain sweeps (**B**). At the ratio of 1.3:1, the network showed G′ = 10,000 Pa and critical yield stress at 10% strain. Meanwhile, the 2.6:1 network showed G′ = 1000 Pa and no yield up to 100% strain. The angular frequency SD values of G′ and G″ are 6.02% and 4.01% for the 1.3:1 ratio, and 9.7% and 7.09% for the 2.6:1 ratio, respectively. The strain SD values of G′ and G″ are 19.2% and 8.09% for the 1.3:1 ratio, and 8.39% and 9.21% for the 2.6:1 ratio, respectively.

**Figure 4 gels-09-00481-f004:**
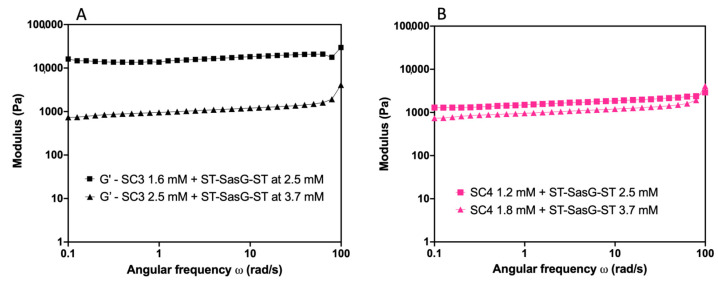
Viscoelastic changes in response to the increase in the total protein concentration. (**A**) Different SC3 + ST–SasG–ST combinations at 1.6 mM + 2.5 mM (black squares) and 2.5 mM + 3.7 mM (black triangles). SC3 shows G′ = 10,000 Pa at lower protein concentrations and G′ = 1000 Pa at higher protein concentrations. SD values are 3.93% and 4.05%, respectively. (**B**) Different SC4 + ST–SasG–ST combinations of 1.2 mM + 2.5 mM (pink squares) and 1.8 mM + 3.7 mM (pink triangles). SC4 shows G′ = 1000 Pa in both protein concentrations. SD values are 1.01% and 2.36%, respectively. Protein concentration was increased by 1.5 at each step.

**Table 1 gels-09-00481-t001:** Composition of the hydrogels based on the molar content of ST and SC.

	ST				
	1 mM (2 ST Units)	2 mM (4 ST Units)	3 mM (6 ST Units)	3.5 mM (7 ST Units)	4 mM (8 ST Units)
1 mM SC3 (3 SC units)	0.6:1	1.3:1	2:1	2.3:1	2.6:1
1 mM SC4 (4 SC units)	0.5:1	1:1	1.5:1	1.75:1	2:1

## Data Availability

All data is available in the Excel file provided with the Appendix A of this article.

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
