# Peer review of "Modulating the Viscoelastic Properties of Covalently Crosslinked Protein Hydrogels"

_gels, 2023, doi:10.3390/gels9060481_

Round 1

Reviewer 1 Report

The manuscript entitled, ‘Modulating the viscoelastic properties of covalently crosslinked protein
hydrogels’ reported protein based hydrogels and its viscoelastic nature. The article could be published
before accounting the following comments.

1. The gels block length is in nanometer as of the schematic. Did the author find about the porosity?

2. The swelling in water in not given also. That should be given also.

3. The rheology plot shows distinct gel strength. What is the driving force to form such high strength
materials? Explanation needed.

4. Some article with significance could have been studied for better literature review: Ganguly, S., Das,
P., & Das, N. C. (2020). Characterization tools and techniques of hydrogels. In Hydrogels based on
natural polymers (pp. 481-517); Alkekhia, D., LaRose, C., & Shukla, A. (2022). β-Lactamase-
Responsive hydrogel drug delivery platform for bacteria-triggered cargo release. ACS Applied
Materials & Interfaces, 14(24), 27538-27550.

No comments

Author Response

The manuscript entitled, ‘Modulating the viscoelastic properties of covalently crosslinked protein hydrogels’ reported protein based hydrogels and its viscoelastic nature. The article could be published before accounting the following comments.

  1. The gels block length is in nanometer as of the schematic. Did the author find about the porosity?

For our proposed application, cell culture, we would encapsulate the cells in the gels before initiating gelation. The key molecules that need to diffuse through the gels are oxygen and small molecule nutrients (such as glucose), which would be able to readily diffuse through even nanometer sized pores We therefore concluded that it was not useful at this point to pursue any measures of pore size. Moreover, as we are sure the reviewer appreciates, making porosity measurements is fraught with the potential for artifacts, and there is no generally agreed upon way to make such measurements.

  1. The swelling in water in not given also. That should be given also.

In the revised manuscript, we have added swelling data, as requested by the reviewer. The swelling is in revised Figure SI1.

  1. The rheology plot shows distinct gel strength. What is the driving force to form such high strength materials? Explanation needed.

These are chemically cross-linked gels, held together by covalent bonds, which form spontaneously between the two components upon mixing. It is a covalently cross-linked gel, which would be the origin of its strength. We have added a text in the revised manuscript to briefly explain this point.

  1. Some article with significance could have been studied for better literature review: Ganguly, S., Das, P., & Das, N. C. (2020). Characterization tools and techniques of hydrogels. In Hydrogels based on natural polymers (pp. 481-517); Alkekhia, D., LaRose, C., & Shukla, A. (2022). β-Lactamase-Responsive hydrogel drug delivery platform for bacteria-triggered cargo release. ACS Applied Materials & Interfaces, 14(24), 27538-27550.

We thank the reviewer for bringing these papers to our attention. We have added the Ganguly, S., Das, P., & Das, N. C. (2020). Characterization tools and techniques of hydrogels. In Hydrogels based on natural polymers (pp. 481-517) citation to our paper to improve our introduction as it is a great review of hydrogels characterisation techniques. However, we would like to differentiate between recombinantly produced' protein-based hydrogels, and hydrogels that use an abundant protein - eg alginate or collagen. The β-Lactamase-Responsive hydrogel was not produced recombinantly, and as such, we do not think that it is a relevant paper for our work.

Reviewer 2 Report

In the manuscript titled ¨Modulating the viscoelastic properties of covalently cross-linked protein hydrogels¨, the authors declare ¨investigated how the identity of the protein pairs, the molar ratio of ST:SC, and the concentration of the proteins influence the viscoelastic response of the hydrogels. By showing tuneable changes in protein hydrogel rheology, we increased the capabilities of synthetic biology to create novel materials, allowing engineering biology to interface with soft matter, tissue engineering, and material science¨. Nevertheless, many consideration needs to be solved before the acceptance of this manuscript.

1. The authors include a small introduction section missing information about the SpyTag - SpyCatcher systems, for example, the origin of the bacteria where this protein is isolated. Considering the function of the introduction, this basic information needs to be included here, not in the discussion section. Please modify the introduction. Also, in the introduction section, the authors declare  ¨We have shown that ST and SC can be successfully used to form biocompatible covalent molecular networks with distinct physical properties that allow encapsulation of mammalian cells without loss of cell viability (unpublished manuscript)¨. These mentioned results are crucial to demonstrate the usage of the proposed hydrogel but aren't presented in this manuscript.  As a consequence, this manuscript looks like a part of another investigation. Encouraging the authors to avoid including unpublished results or personal communications is relevant. 

2. In the material and methods sections, the authors declare that bacterial strains and plasmids used in this study are listed in the Supplementary material. Nevertheless, the supporting document (excel) does not present the corresponding list. The loss of this information or appropriated references diminishes the quality of the manuscript.

3. In the material and methods sections, the authors are missing to declare a) the method used for analyzing the recombinant protein production (i.e., SDS-PAGE or WB), recombinant protein purity, and concentration. This is relevant because every batch of recombinant protein needs characterization. Please include this information. b) The ratio of the SC- ST in each construction, they declared in the results sections, but following the scientific method, this information needs to be included in this section. Also, gelation conditions need improvement, I. e., room temperature. Do the lyophilized proteins make the hydrogel without adding water or buffer? Please declare the specific conditions needed for gelation (i.e., room temperature, buffer addition, pH, etc.). c) Please describe the formulated hydrogel, i.e., size and dimensions, and include a picture in the result section. d) Declares the number of hydrogels used for the rheological analysis. This information is essential for statistical analysis.

4. In the results section, the authors declare as a reference that ¨The structure of SasG (GEG) .... and longer SasG arrays, .... have been characterised in solution by small angle x-ray light scattering (SAXS) and atomic force microscopy (AFM) [9].¨ But, in this manuscript, the author declares  ¨... a longer version of the crosslinker, which we named SasGlong¨ and the AFM analysis is missing. Please include the AFM characterization of SasGlong. Considering the apparent limitation of the methods 

5. The author indicates Supplemental Figures (i.e., FigS8), but the document does not include figures (excel). The supplemental only consists of the measurements obtained directly by the equipment. 

6. The results demonstrated here are not sufficient to sustain the conclusion proposed by the author, considering that the rheological results are informative and it is necessary for the structural characterization data (i.e., microscopy and scattering) to generate an integrative (i.e., hydrogel-cel interaction) acknowledgment of the hydrogel properties. Also extensive mechanical and morphological characterization is needed. 

Currently, the document looks like a part of another complete manuscript. The author's approximation of based a manuscript solely on the rheological analysis of a hydrogel is a limited approach that lacks novelty.

Author Response

In the manuscript titled ¨Modulating the viscoelastic properties of covalently cross-linked protein hydrogels¨, the authors declare ¨investigated how the identity of the protein pairs, the molar ratio of ST:SC, and the concentration of the proteins influence the viscoelastic response of the hydrogels. By showing tuneable changes in protein hydrogel rheology, we increased the capabilities of synthetic biology to create novel materials, allowing engineering biology to interface with soft matter, tissue engineering, and material science¨. Nevertheless, many consideration needs to be solved before the acceptance of this manuscript.

  1. The authors include a small introduction section missing information about the SpyTag - SpyCatcher systems, for example, the origin of the bacteria where this protein is isolated. Considering the function of the introduction, this basic information needs to be included here, not in the discussion section.

We have made clear in the introduction what these proteins are – and have referenced appropriately. We have also expanded the introduction and added more details on the SpyTag-SpyCatcher system.  

 Please modify the introduction. Also, in the introduction section, the authors declare  ¨We have shown that ST and SC can be successfully used to form biocompatible covalent molecular networks with distinct physical properties that allow encapsulation of mammalian cells without loss of cell viability¨. These mentioned results are crucial to demonstrate the usage of the proposed hydrogel but aren't presented in this manuscript. 

This paper focuses on extensive characterisation of the physical properties of these hydrogels, and how those properties change as the composition of the gels change. Such focus is essential, to allow others to determine if this system could be useful for their application. We mention that we are able to encapsulate mammalian cells in the gels without loss of viability, to let readers know that this application is possible (manuscript submitted). This paper, however, is about the physical properties of the gels, and not their use in mammalian cell growth.  

  1. In the material and methods sections, the authors declare that bacterial strains and plasmids used in this study are listed in the Supplementary material. Nevertheless, the supporting document (excel) does not present the corresponding list. The loss of this information or appropriated references diminishes the quality of the manuscript.

The bacterial strains and plasmids used in this study are listed in the Supplementary material that was uploaded during submission. The excel sheet only contains the raw data from our experiments for research transparency.

  1. In the material and methods sections, the authors are missing to declare a) the method used for analyzing the recombinant protein production (i.e., SDS-PAGE or WB), recombinant protein purity, and concentration. This is relevant because every batch of recombinant protein needs characterization. Please include this information.]

Each batch of protein was prepared in the same way, and analysed by Coomassie Blue staining of an SDS-PAGE. We observed no significant variation between batches. We added a sentence highlighting the use of SDS-page. We estimate that the proteins are ~90% pure.

  1. b) The ratio of the SC- ST in each construction, they declared in the results sections, but following the scientific method, this information needs to be included in this section.

In the original manuscript we stated that we use 3 repeats per gel in the materials and methods. We also stated in materials and methods how we determined the concentration of components (A280 and extinction coefficient calculated from the sequence – that we provided in the SI)

Also, gelation conditions need improvement, I. e., room temperature. Do the lyophilized proteins make the hydrogel without adding water or buffer? Please declare the specific conditions needed for gelation (i.e., room temperature, buffer addition, pH, etc.).

The gels form easily and we have not observed sensitivity to temperature, buffer, or pH. We stated the conditions we used, but have added the statement that we have not observed a great sensitivity to conditions (and non-was reported by Howarth for the SC-ST reaction in solution – as we referenced). We state the concentration of the protein in solution. We thought that it was therefore obvious that the lyophilised proteins are dissolved in water or buffer pre-mixing, however we clarified this in the main text. Moreover, in the materials and methods section of the original manuscript we stated “Lyophilised proteins were individually dissolved in distilled water to give stock solutions of the desired concentration”. We have also expanded on the SpyTag-SpyCatcher system highlighting its ease of use due to spontaneous gelation.

  1. c) Please describe the formulated hydrogel, i.e., size and dimensions, and include a picture in the result section.

The gel is mouldable. Depending on how much protein solution we mix, a different sized gel can be made. We have added a photo of our mouldable hydrogels as an insert in Fig. 2. The size of the hydrogel depends on the amount of protein that is used to make it. For the photo we have added, the hydrogel was made from ~ 50 uL of protein solutions.

  1. d) Declares the number of hydrogels used for the rheological analysis. This information is essential for statistical analysis.

In the original manuscript we stated that we made 3 repeats of the rheology measurement for each gel in the materials and methods section.

  1. In the results section, the authors declare as a reference that ¨The structure of SasG (GEG) .... and longer SasG arrays, .... have been characterised in solution by small angle x-ray light scattering (SAXS) and atomic force microscopy (AFM) [9].¨ But, in this manuscript, the author declares  ¨... a longer version of the crosslinker, which we named SasGlong¨ and the AFM analysis is missing. Please include the AFM characterization of SasGlong. Considering the apparent limitation of the methods 

The SasGlong analysis is included in Gruszka 2015, we have highlighted this reference as published work in the text.

  1. The author indicates Supplemental Figures (i.e., FigS8), but the document does not include figures (excel). The supplemental only consists of the measurements obtained directly by the equipment. 

We believe that this reviewer did not see the supplemental file that was uploaded, but only saw the raw file data. We checked and it is there.

  1. The results demonstrated here are not sufficient to sustain the conclusion proposed by the author, considering that the rheological results are informative and it is necessary for the structural characterization data (i.e., microscopy and scattering) to generate an integrative (i.e., hydrogel-cel interaction) acknowledgment of the hydrogel properties. Also extensive mechanical and morphological characterization is needed. 

Currently, the document looks like a part of another complete manuscript. The author's approximation of based a manuscript solely on the rheological analysis of a hydrogel is a limited approach that lacks novelty.

Unfortunately, we find it hard to understand what the reviewer is saying or asking for in this paragraph. The paper includes extensive mechanical characterisations – indeed it is focused on the rheological properties of the the hydrogels, and how they change in response to changing the components of the gels. 

From these comments, it is not clear to us what microscopy or scattering data the reviewer would like to see included in the gel. Most importantly, we do not think that there are ‘microscopy or scattering’ that would be relevant to enhancing the characterisations that we present.  

We find it hard to respond to these comments, which contradict the other reviewers’ assessment of the work.  For example, reviewer three recommends, “In the introduction section, the authors should point out the novelty and importance of this work.” – which we have done. 

Reviewer 3 Report

In this manuscript, engineered proteins to form covalent molecular networks with defined physical characteristics were designed. The viscoelastic response of the hydrogels was influenced by SpyTag (ST) peptide and SpyCatcher (SC) protein. By showing tuneable changes in protein hydrogel rheology, the capabilities of synthetic biology to create novel materials may increase. I suggest the publication of this manuscript in Gels if the author can address the following concerns.

1. In the introduction section, the authors should point out the novelty and importance of this work.

2. The preparation process of the ST:SC hydrogels should be more specific.

3. What is the swelling behavior of protein gels with different SC:ST ratios? The swelling characteristic of hydrogel is important, and it also influences viscoelastic properties.

4. One recently published paper (DOI: 10.1002/adfm.202207388) indicated that the molecular interaction influenced the viscoelastic property of the hydrogel. Which kind of molecular interactions were included in this hydrogel? The author should point out how the viscoelastic property influenced the specific molecular interactions, and compare this manuscript with the above paper regarding the molecular interactions. Generally, the viscoelastic property influenced the rheology of the hydrogel, thus, the viscoelastic feature was explored more for 3D printing. However, the author did not show anything about 3D printing. Why did the author think viscoelasticity was important for these protein hydrogels? Besides, did the polymerization occur during the gelation? If not, why the author thought this hydrogel was based on the covalently crosslinked strategy?

5. The degree of crosslinking should be characterized since the authors indicated that “the presence of an overabundance of SCs limits the number of crosslinks that can occur and the STs are ‘blocked’ from cross-linking.”

6. The strain sweeps of SC4 combined with ST-SasG-ST and ST-SasGlong-ST should also be shown.

Author Response

In this manuscript, engineered proteins to form covalent molecular networks with defined physical characteristics were designed. The viscoelastic response of the hydrogels was influenced by SpyTag (ST) peptide and SpyCatcher (SC) protein. By showing tuneable changes in protein hydrogel rheology, the capabilities of synthetic biology to create novel materials may increase. I suggest the publication of this manuscript in Gels if the author can address the following concerns.

  1. In the introduction section, the authors should point out the novelty and importance of this work.

We thank the reviewer for this comment. We have improved our introduction stressing the importance and the novelty of the work.

  1. The preparation process of the ST:SC hydrogels should be more specific.

We have revised our material and methods section to clarify. However, the preparation process is as simple as we portray it - mix the two components at room temperature and they spontaneously form gels (because the spontaneously form covalent cross-links). The simplicity of gel formation is one of the key features of this system. Gel preparation really is as simple as mixing the two components in the desired amounts. We have added two references in the main text where the authors created hydrogels using the SpyTag-SpyCatcher system in the same way we did, highlighting the spontaneous gelation and the ease of the method.

  1. What is the swelling behavior of protein gels with different SC:ST ratios? The swelling characteristic of hydrogel is important, and it also influences viscoelastic properties.

In the revised manuscript, we have added the swelling behavior at different SC:ST ratios as suggested by the reviewer.

  1. One recently published paper (DOI: 10.1002/adfm.202207388) indicated that the molecular interaction influenced the viscoelastic property of the hydrogel. Which kind of molecular interactions were included in this hydrogel? The author should point out how the viscoelastic property influenced the specific molecular interactions, and compare this manuscript with the above paper regarding the molecular interactions. Generally, the viscoelastic property influenced the rheology of the hydrogel, thus, the viscoelastic feature was explored more for 3D printing. However, the author did not show anything about 3D printing. Why did the author think viscoelasticity was important for these protein hydrogels? Besides, did the polymerization occur during the gelation? If not, why the author thought this hydrogel was based on the covalently crosslinked strategy?

We specified that the molecular interactions in the hydrogels are the covalent cross-links. Those are the only ones we know about and which we can influence by changing designs. We know that the proteins must be folded, because if we denature the proteins with urea, after covalent bond formation, the properties of the gel change. We have added this clarification in the main text. We have no indication that any other molecular interactions are important. We have carried out additional experiments above the 4 mM concentration (ST-SasG-ST = 6 mM), but we realized that the viscosity of the solution was too high and the viscoelastic properties of the hydrogels were reduced. Therefore, we did not present any data above 4 mM. Regarding the viscoelastic properties, our primary concern is because we seek to use these gels as scaffolds for cell growth & proliferation. We therefore want their viscoelastic properties to be well-matched to those of the tissues we seek to use them with. We have added a sentence in the text highlighting this. Finally, we have also specified in the text our requirement for 3D printing and how the viscoelastic properties of the materials are critical to successful biofabrication.

  1. The degree of crosslinking should be characterized since the authors indicated that “the presence of an overabundance of SCs limits the number of crosslinks that can occur and the STs are ‘blocked’ from cross-linking.”

We have removed the sentence on the degree of crosslinking because it is extremely difficult to determine with meaningful accuracy. We have expanded the discussion by adding references to other papers where similar phenomena are reported and the degree of crosslinking is not quantified.

  1. The strain sweeps of SC4 combined with ST-SasG-ST and ST-SasGlong-ST should also be shown.

The strain sweeps of SC4 combined with ST-SasG-ST and ST-SasGlong-ST are already present in the SI of the manuscript we originally submitted.

Reviewer 4 Report

In this work Boni * et al. designed and programmed engineered proteins to form covalent molecular networks with defined physical characteristics for tailored engineering applications.

Their hydrogel design incorporates the SpyTag peptide and SpyCatcher protein that spontaneously form covalent crosslinks upon mixing.

The authors are presenting interesting study and are very ambitious.

However, there are still few minor text revision / correction:

The part of the conclusions is included in the discussions without specifying that it also contains the conclusions.

The part of the conclusions is to be set apart or to specify where the conclusions are.

Round 2

Reviewer 3 Report

I suggest the acceptance of this manuscript.

Additional English revision and improvement should be implemented.

Author Response

We thank this reviewer for acknowledging our work in addressing their comments and for stating that our paper is now worthy of publication with Gels.